# Simple Predictors for Cardiac Fibrosis in Patients with Type 2 Diabetes Mellitus: The Role of Circulating Biomarkers and Pulse Wave Velocity

**DOI:** 10.3390/jcm11102843

**Published:** 2022-05-18

**Authors:** Ekaterina B. Luneva, Anastasia A. Vasileva, Elena V. Karelkina, Maria A. Boyarinova, Evgeny N. Mikhaylov, Anton V. Ryzhkov, Alina Y. Babenko, Alexandra O. Konradi, Olga M. Moiseeva

**Affiliations:** Almazov National Medical Research Centre, 197341 Saint-Petersburg, Russia; vasileva_aa@almazovcentre.ru (A.A.V.); karelkina_ev@almazovcentre.ru (E.V.K.); boyarinova_ma@almazovcentre.ru (M.A.B.); e.mikhaylov@almazovcentre.ru (E.N.M.); ryzhkov_av@almazovcentre.ru (A.V.R.); babenko_ayu@almazovcentre.ru (A.Y.B.); konradi_ao@almazovcentre.ru (A.O.K.); moiseeva_om@almazovcentre.ru (O.M.M.)

**Keywords:** cardiac fibrosis, diabetes mellitus, pulse wave velocity

## Abstract

Cardiac fibrosis is the basis of structural and functional disorders in patients with diabetes mellitus (T2DM). A wide range of laboratory and instrumental methods is used for its prediction. The study aimed to identify simple predictors of cardiac fibrosis in patients with T2DM based on the analysis of circulating fibrosis biomarkers and arterial stiffness. The study included patients with T2DM (*n* = 37) and cardiovascular risk factors (RF, *n* = 27) who underwent ECHO, cardiac magnetic resonance imaging (MRI), pulse wave analysis (PWV), reactive hyperemia (RH), peripheral arterial tonometry, carotid ultrasonography, and assessment of serum fibrosis biomarkers. As a control group, 15 healthy subjects were examined. Left ventricular concentric hypertrophy was accompanied by an increased serum galectin-3 level in T2DM patients. There was a relationship between the PICP and HbA1c levels in both main groups (R2 = 0.309; *p* = 0.014). A negative correlation between PICP level and the global longitudinal strain (GLS) was found (r = −0.467; *p* = 0.004). The RH index had a negative correlation with the duration of diabetes (r = −0.356; *p* = 0.03), the carotid-femoral PWV (r = −0.371; *p* = 0.024), and the carotid intima-media thickness (r = −0.622; *p* < 0.001). The late gadolinium-enhanced (LGE) cardiac MRI was detected in 22 (59.5%) T2DM and in 4 (14.85%) RF patients. Diabetes, its baseline treatment with metformin, HbA1c and serum TIMP-1 levels, and left ventricle hypertrophy had moderate positive correlations with LGE findings (*p* < 0.05). Using the multivariate regression analysis, increased TIMP-1 level was identified as an independent factor associated with cardiac fibrosis.

## 1. Introduction

Cardiovascular (CV) complications remain the leading cause of premature death and disability in type 2 diabetes mellitus (T2DM) [1]. According to the population-based studies, patients with T2DM have a 2–5-fold increased CV risk when combined with traditional risk factors such as hypertension, dyslipidemia, advanced age, obesity, and smoking [2]. At the same time, obesity is among the strongest predictors of T2DM development. T2DM promotes pro-inflammatory and prothrombotic signaling, resulting in endothelial dysfunction and atherogenesis acceleration associated with CV events [3].

Heart failure seems to become one of the most prevalent and serious T2DM consequences, being considered either manifestation of diabetic cardiomyopathy or macrovascular ischemic heart disease or both [4]. Left ventricular (LV) hypertrophy with myocardial fibrosis is a typical sign of diabetic cardiomyopathy. Echocardiographic (ECHO) parameters of LV diastolic function and global longitudinal strain (GLS) are widely used as nonspecific surrogate markers of myocardial fibrosis in clinical practice [5,6]. Unlike ECHO as a screening tool for functional LV assessment, cardiac magnetic resonance imaging (MRI) is a robust non-invasive method for myocardial fibrosis detection and quantification. Despite cardiac MRI’s potential benefits, its real implementation is limited by low availability and high cost [7]. Plasma-circulating biomarkers are also widely used for indirect cardiac fibrosis assessment; however, their diagnostic value is still a matter of debate [8,9]. T2DM is one of the major determinants of accelerated arterial stiffening along with hypertension and age. It is suggested that cardiac fibrosis in T2DM patients is associated with increased arterial wall stiffness as well. The increased arterial stiffness has been shown significantly impact LV afterload and, therefore, is crucial for the development of heart failure with preserved ejection fraction (HFpEF) [10,11].

The elaboration of non-invasive markers predicting cardiac fibrosis is of essential importance since fibrosis is strongly associated with CV events and may require more aggressive treatment.

The present study aimed at identifying simple predictors of cardiac fibrosis in patients with T2DM based on the analysis of circulating fibrosis biomarkers and arterial stiffness. T2DM has been shown to be associated with tissue fibrosis in general and cardiac fibrosis in particular [12]. Plasma concentrations of circulating biomarkers that may characterize the presence and extent of fibrosis are associated with other morbidity and risk factors, such as obesity and hypertension. Moreover, their reference level should be evaluated in healthy subjects for the assessment of their significance when changed. Therefore, along with T2DM patients, we included two more subgroups: subjects without T2DM but with cardiovascular risk factors and healthy controls.

## 2. Materials and Methods

### 2.1. Study Population

The cross-sectional study recruited subjects from the outpatient clinic of the Almazov National Medical Research Centre between August 2019 and July 2020. Subjects fulfilling inclusion criteria were invited into the study by a treating physician (screening) and referred to an investigator. The subjects were divided into three groups: T2DM patients, patients with CV risk factors (RF), and healthy control (HC) subjects.

The inclusion criteria for the T2DM group were the following: glycated hemoglobin (HbA1c) level > 6.5% at screening and T2DM diagnosed >1 year ago. The RF group inclusion criteria were the combination of two common risk factors: obesity (BMI > 30.0 kg/m^2^) and hypertension (office blood pressure level > 140/90 mm Hg) or dyslipidemia (the history of LDL cholesterol > 3 mmol/L). The HC group comprised blood donors without a history of CV disease.

The exclusion criteria were changes in pharmacological treatment (drugs and/or doses) within 1 month; inadequate blood pressure control (≥140/80 mm Hg at office visits); a history of coronary artery disease, myocardial infarction, or TIA/stroke; LV ejection fraction < 50%; an implanted pacemaker or cardioverter-defibrillator; ongoing infectious or neoplastic diseases; documented osteoporosis or osteopenia; pregnancy or breastfeeding; any intervention or surgery within 6 months. The study was approved by the Ethics Committee of the Almazov Centre (No. 05072019 dated 8 July 2019), and all participants signed an informed consent form before the inclusion. The baseline evaluation included medical history and physical examination, routine laboratory tests, circulating fibrosis biomarkers, endothelial function assessment, pulse wave analysis, carotid intima-media thickness, and echocardiography. Contrast-enhanced cardiac MRI was performed in the T2DM and RF groups. The HC group underwent biomarker analysis only.

The primary study assessment measure included the evaluation of a possible association between cardiac fibrosis as detected by cardiac MRI, artery stiffness, and circulating biomarkers. The secondary study analysis was the evaluation of factors independently associated with the presence of cardiac fibrosis.

This observational study was registered as a part of an umbrella project #075-15-2020-800 by the Ministry of Science and Higher Education. The local legislation does not require observational studies registration in public databases.

### 2.2. Blood Assays

Peripheral venous blood samples were obtained at the first visit. Serum samples were obtained following centrifugation at 2500× *g* for 10 min at 4 °C. Samples were aliquoted and stored at −80 °C until required. Lab parameters included HbA1c (Tina-Quant Hemoglobin A1c Gen.3, Cobas Integra 400+, Roche Diagnostics GmbH, Mannheim, Germany), creatinine, lipids (Cobas Integra 400+, Roche Diagnostics GmbH, Mannheim, Germany), high-sensitivity C-reactive protein by the immunoturbidimetric CRP-Latex assay (Tina-quant^®^ CRP latex, Cobas Integra 400+, Roche Diagnostics GmbH, Mannheim, Germany), NT-proBNP (Elecsys, Roche Diagnostics GmbH, Mannheim, Germany), and soluble suppression of tumorigenicity 2 sST2 (Presage ST2 kit, Critical Diagnostics, CA, USA). Carboxy-terminal propeptide of collagen 1 (PICP, USCN Life Science, Wuhan, China), amino-terminal propeptide of collagen 3 (PIIINP, USCN Life Science, Wuhan, China), carboxy-terminal telopeptide of collagen 1 (ICTP, MyBioSource, San Diego, CA, USA), transforming growth factor β-1 (TGFβ1, R&D systems Inc., Minneapolis, MN, USA), matrix metalloproteinase 9 (MMP9, R&D systems Inc., Minneapolis, MN, USA), tissue inhibitor of metalloproteinase 1 (TIMP1, R&D systems Inc., Minneapolis, MN, USA), and galectin-3 (R&D systems Inc., Minneapolis, MN, USA) were quantified using a specific enzyme-linked immunosorbent assay (ELISA, microplate reader “Bio-Rad 680”, Bio-Rad Laboratories Inc, Hercules, California, USA) as a serum biomarker of fibrosis. These biomarkers were selected based on previous publications demonstrating their role in cardiac fibrosis [8].

### 2.3. Blood Pressure Measurement

Office blood pressure (BP) was measured with a calibrated automatic sphygmomanometer (OMRON M3 Expert, Omron Dalian, Kioto, Japan). We used a BP cuff that fits the participants’ arm circumference. Three measurements were performed in a seated position after a 5-min rest with a 5-min interval. The average value of the two last measurements was calculated.

### 2.4. Pulse Wave Analysis

BP waveforms were recorded on the carotid and femoral arteries using applanation tonometry (SphygmoCor, AtCor Medical, Sidney, Australia) in standardized conditions (supine position, quiet atmosphere, and temperature 24 °C). Caffeine and smoking were not allowed within 3 h before evaluation. Pulse wave velocity (PWV) was calculated automatically according to the patient’s height, weight, and brachial BP assessed before the procedure. The cut-off value for carotid-femoral PWV was 10 m/s [13].

### 2.5. Reactive Hyperemia Peripheral Arterial Tonometry

Endothelial function was assessed using peripheral arterial tonometry with the Endo-PAT2000 device (Itamar Medical, Caesarea, Israel). Reactive hyperemia index (RHI) was evaluated according to the previously reported protocol [14]. RHI < 1.67 was considered a sign of peripheral arterial endothelial dysfunction [14].

### 2.6. Echocardiography

Echocardiography was performed using the Vivid 7 system (GE Healthcare, Chicago, IL, USA) according to a standard protocol with an assessment of global longitudinal strain (GLS) and the ratio of early diastolic transmitral flow velocity to the average peak early diastolic mitral annular velocity (E/e’) as a measure of filling pressures [14,15]. LV mass/body surface area >115 g/m^2^ in men and >95 g/m^2^ in women was defined as LV hypertrophy [15,16].

### 2.7. Cardiac MRI

Cardiac MRI was carried out using a high-field 3 T MRI scanner MAGNETOM Trio A Tim System 3T (Siemens Healthineers, Erlangen, Germany) in an ECG-synchronized mode. The procedure was performed according to the standard protocol, which included late gadolinium enhancement (LGE) sequences using PSIR (Phase-sensitive Inversion Recovery) sequences with an inversion time of 200 ms, a repetition time of 8.5 ms, and an echo time of 3.5 ms, after 10 min gadopentetate dimeglumine (0.2 mmol/kg, Gadovist, BayerHealthcare, Berlin, Germany) administration. All analyses were performed by an independent reader. Left ventricular function was evaluated semi-automatically using commercially available software (Syngo Via, Siemens Healthineers, Erlangen, Germany) according to ACCF/ACR/AHA/NASCI/SCMR recommendations [17]. Following automatic contour detection of the LV endocardium, all borders were corrected manually. The extent of LGE was calculated semi-quantitatively by counting the number of LV segments showing visually-determined LGE. LGE volume was calculated by summation of the LGE areas in all short-axis slices, which was expressed as a volumetric proportion of the total LV myocardium using a similar approach previously described [18]. Analysis of LGE was performed visually in a short-axis stack and a four-chamber view for the presence of LGE using the 17-segment model of the American Heart Association (AHA) [19].

### 2.8. Carotid Ultrasonography

Carotid ultrasound studies were performed by high-resolution B-mode ultrasonography (Vivid7, GE Healthcare, Chicago, IL, USA) with a linear array broadband transducer 7 MHz. The standard protocol included bilateral measurements at a distance of 1 cm from the bifurcation of the common carotid artery along its posterior wall in three positions (anterior, middle, and posterior longitudinal). The intima-media thickness (IMT) was defined as the distance between the first and second echogenic lines of the artery. Then, the mean IMT on both sides was calculated as an arithmetic mean of three dimensions. The subclinical vascular damage was detected if IMT ≥ 0.9 mm.

### 2.9. Statistical Analysis

Data are presented as mean (±standard deviation) or median (interquartile range) for normal and abnormal distributed continuous variables, respectively, whereas categorical data were expressed as frequencies and percentages. Differences in baseline characteristics were evaluated using Student’s *t*-test, Mann–Whitney U test, or Chi-square test, depending on the variable category. The one-way ANOVA and post hoc (Tukey–Kramer test) were also used for the comparison of parameters in three groups. Spearman correlation was used to evaluate relationships involving ordinal variables. Statistical significance was considered at *p* < 0.05. A correlation matrix incorporating all evaluated clinical, laboratory parameters, and serum biomarkers was created. Those factors that had a statistically significant correlation with MRI-LGE-positive findings were studied using logistic regression. When factors had a significant cross-correlation (ρ > 0.65), one of them was selected for the multivariate regression analysis based on a higher correlation coefficient with LGE positivity. Factors independently associated with MRI-detected fibrosis were evaluated using two multivariate binary logistic regression models: the first aimed at the inclusion of all factors significantly correlated with MRI-detected fibrosis and incorporated a combination of clinical, echocardiography, and biochemical markers; the second model aimed at the inclusion of only additional factors with moderate-to-significant correlation (serum biomarkers and pulse wave velocity). We suggested that the latter model would be practically applicable for the identification of cardiac-fibrosis-positive serum biomarkers in the mixed population. The regression analysis was performed for the total study population and for T2DM patients separately. IBM software SPSS Version 23 (SPSS, Inc., Chicago, IL, USA) and Statistica 13.0 (Statsoft, Tulsa, OK, USA) were applied for statistical analysis.

## 3. Results

### 3.1. Patient Baseline Characteristics

The study population comprised 79 subjects: 37 subjects in the T2DM group (age ranging between 44 and 70 years), 27 subjects in the risk factors (RF) group (age 40–68 years), and 15 subjects in the healthy control (HC) group (age 50–65 years). The baseline subjects’ characteristics are summarized in Table 1. There were no differences in age, sex, blood pressure (BP) level, and smoking status between the T2DM and RF groups. Patients without diabetes tended to have a higher body mass index (BMI) with the same waist circumference, but statistically it was not significant. All hypertensive patients were receiving angiotensin-converting enzyme inhibitors (ACEI) or angiotensin II receptor blockers (ARB). Although statistically borderline, the prevalence of carotid intima-media thickness (IMT) ≥ 0.9 mm tended to be higher in T2DM patients (19% vs. 3.7% in the RF group, *p* = 0.052), whereas the carotid-femoral PWV ≥ 10 m/s was significantly more prevalent in the T2DM group (32% vs. 3.7%, respectively; χ2 = 8.65; *p* = 0,003). Despite a significantly lower reactive hyperemia index in the diabetic patients, the percentage of patients with reactive hyperemia index (RHI) < 1.67 did not differ between the T2DM and RF groups: 65% vs. 52%, respectively (*p* = 0.296). The RHI had a negative correlation with the duration of diabetes (r = −0.356; *p* = 0.03), the carotid-femoral PWV (r = −0.371; *p* = 0.024), and the carotid IMT (r = −0.622; *p* < 0.001). A negative correlation between the RHI and carotid IMT was observed in the RF group (r = −0.558; *p* = 0.002). There was no difference in the estimated glomerular filtration rate (eGFR) between the groups.

### 3.2. Laboratory Measurements

There was a significant difference in serum lipid levels between the groups. Thus, RF patients were characterized by higher mean low-density cholesterol (LDL-C) intima-media thickness and triglyceride levels (Table 2). Surprisingly, the HC subjects had similar lipid levels when compared with the RF group. Serum Carboxy-terminal propeptide of collagen 1 (PICP) and amino-terminal propeptide of collagen 3 (PIIINP) levels as the collagen metabolism markers were significantly increased in the T2DM and RF groups compared to the HC subjects. Among T2DM patients, statin therapy was associated with a lower PICP level: 129 ng/mL [QIR: 115–146] vs. 192 ng/mL [QIR: 169–195] in patients without statins (*p* < 0.001). Interestingly, there were no differences in PIIINP levels between the T2DM and RF groups. Concentrations of matrix metalloproteinase 9 (MMP9), tissue inhibitor of metalloproteinase 1 (TIMP1), and carboxy-terminal telopeptide of collagen 1 (ICTP) were the lowest in the HC group. However, T2DM patients had higher TIMP1 and ICTP levels. The increased soluble suppression of tumorigenicity 2 (sST2) level was associated with an increase in IMT in both T2DM (r = 0.361; *p* = 0.028) and RF patients (r = 0.499; *p* = 0.008), presumably due to negative effects on endothelial function and RHI as its marker (R^2^ = 0.357; *p* = 0.004). Higher serum TGFβ1 levels were revealed in T2DM and RF groups compared to the HC subjects, but serum TGFβ1 had a significant positive correlation with BMI (body mass index) (r = 0.564; *p* = 0.0002), waist circumference (r = 0.432; *p* = 0.008) and a negative correlation with eGFR (r = −0.471; *p* = 0.008) only in the T2DM group. Serum galectin-3 level, as well as TGF, was predominantly increased in T2DM patients. There were no gender differences in the profile of serum fibrosis biomarkers.

### 3.3. Echocardiography and Cardiac MRI Analysis

Left ventricle (LV) hypertrophy was revealed in 30 diabetic (81%) and 9 (33%) RF patients (χ2 = 15.4; *p* = 0.0005), including 25 (67.6%) and 5 (18.5%) patients (*p* = 0.0005) with concentric LV hypertrophy, respectively. The LV concentric hypertrophy was accompanied by an increased serum galectin-3 level in T2DM patients: 9.92 ng/mL [QIR: 8.38–12.96] vs. 8.14 ng/mL [QIR: 6.58–9.85], *p* = 0.039. Left atrial (LA) enlargement (index LA volume ≥ 34 mL/m^2^) was noted in 21 (56.8%) T2DM and 8 (29.6%) RF patients (*p* = 0.094). In T2DM patients, LA enlargement was associated with a higher serum PIIINP level (r = 0.434; *p* = 0.007). A negative correlation between PICP level and the global longitudinal strain (GLS) was found (r = −0.467; *p* = 0.004). This fact is especially interesting since we revealed the relationship between PICP and HbA1c levels in both main groups (R2 = 0.309; *p* = 0.014). Another marker of diastolic dysfunction such as the E/e’ ratio correlated with HbA1c (r = 0.426; *p* = 0.029) and TIMP1 (r = 0.543; *p* = 0.004) levels only in RF patients.

LGE was detected in 22 (59.5%) T2DM patients and in 4 (14.85%) RF patients. LGE was found predominantly in the anteroseptal and inferior mid-wall and basal segments (Figure 1). By semi-quantitative assessment, LGE volume was 13% [QIR: 9–14%] in T2DM patients, while among RF patients only 4% [QIR: 2–4%] (*p* = 0.002).

Diabetes, its baseline treatment with metformin, HbA1c, and serum TIMP-1 levels, and LV hypertrophy had moderate positive correlations with LGE-MRI findings (*p* < 0.05). Although statistically significant, statin treatment, BMI, PWV, and galectin-3 serum level had a weak positive correlation with LGE positivity. LDL-C level had a weak negative correlation with cardiac MRI-detected fibrosis. Univariate logistic regression coefficients are presented in Table 3. The multivariate regression analysis identified that in the first model, TIMP-1 level was the only independent factor associated with cardiac fibrosis. The second model identified TIMP-1 levels and galectin-3 levels as factors independently associated with cardiac fibrosis (Table 4 and Table 5). In T2DM patients, the regression analysis confirmed significant associations of TIMP-1 and PWV with cardiac fibrosis, and the multivariate model identified TIMP-1 as the only factor independently associated with LGE-positive findings (Table 3 and Table 4).

## 4. Discussion

Myocardial fibrosis is usually assessed by an easy scoring system using late gadolinium-enhanced cardiac magnetic resonance (LGE-MRI) imaging. This method is widely used for the risk stratification of patients with cardiovascular disease, yet it is the most accurate method to reveal replacement myocardial fibrosis; however, it is less sensitive in interstitial fibrosis detection. LGE has been reported to predict death and myocardial infarction in a cohort of 1969 patients with and without T2DM [20].

The univariate logistic regression identified the association of T2DM, glycated hemoglobin level, and metformin intake with LGE-positive MRI findings, emphasizing the important role of impaired glucose metabolism in the development of myocardial fibrosis. Previously, an in vitro study has shown that hyperglycemia is a powerful stimulator of fibroblast proliferation, myofibroblast differentiation, and extracellular matrix proteins’ secretion [21].

T2DM patients have a higher prevalence of increased LV mass index, concentric LV hypertrophy in combination with LA enlargement, and increased E/e’ ratio compared with the RF group. This confirms a link between LV hypertrophy with positive LGE and might be associated with poorly controlled T2DM and higher HbA1c levels. LV hypertrophy with diastolic dysfunction is typical for diabetic cardiomyopathy but is also widely prevalent in the elderly population, females, and patients with hypertension and obesity [22,23]. Importantly, we have not found any association between cardiac remodeling and the above-mentioned risk factors in T2DM patients and suggest that this could be explained in part by limited sample size and previous antihypertensive therapy. Moreover, T2DM and RF groups included obese patients, and obesity is one of the pivotal contributors to myocardial fibrosis [24]. This fact is confirmed by the results of the large Multi-Ethnic Study of Atherosclerosis study, where an increased BMI has been shown to be associated with the concentric hypertrophy by cardiac MRI [25].

Pulse wave velocity (PWV) is widely used for arterial stiffness measurement [26]. According to the univariate analysis, cardiac fibrosis is associated with TIMP and galectin-3 levels, as well as with the carotid-femoral PWV. Adjusted for age and blood pressure, T2DM duration appears to be the most important contributor to arterial stiffness [27]. The relationship between arterial stiffness and the severity of LV diastolic dysfunction has been confirmed in a wide variety of cardiovascular diseases [28,29]. Recently, PWV has been linked to different cardiovascular events, including CV mortality [30]. The association between obesity and arterial stiffness confirms the results of the previous study by Desamericq et al. [31]. The correlation between positive LGE and PWV observed in our study supports the conception of common pathophysiological mechanisms of cardiac and vascular remodeling in T2DM. However, according to the multivariate analysis, PWV is not an independent predictor of cardiac fibrosis, while circulating TIMP1 and galectin-3 are strongly associated with cardiac fibrosis being active participants in the pathophysiology of heart and vascular remodeling.

Studies on galectin-3, a protein of the lectin family secreted by activated macrophages and fibroblasts, open novel opportunities for non-invasive cardiac remodeling monitoring in T2DM patients. Previous studies have identified higher circulating galectin-3 levels predicting the onset of HFpEF [32,33]. In addition to LGE-MRI, galectin-3 seems to play an important role in the sudden death risk stratification of heart failure patients [34]. Our study confirms the predictive value of galectin-3 in the diagnosis of cardiac fibrosis.

The circulating serum biomarkers of collagen synthesis and degradation are used for indirect myocardial fibrosis assessment [8]. A distinctive feature of T2DM and RF patients included in this study is the high PIIINP level (as a marker of III collagen synthesis activation) [35]. Circulating PIIINP can serve as a marker of large vessel remodeling [36]. Histological studies have shown increased PICP and PIIINP levels associated with interstitial and perivascular cardiac fibrosis in T2DM, regardless of the presence of coronary atherosclerosis and hypertension [37]. In our study, elevated serum PIIINP levels were associated with LA enlargement as a marker of LV diastolic dysfunction. Opposite, a decrease in the GLS as an early marker of LV systolic dysfunction was related to the circulating PICP level. An interesting finding is a reduction in PICP level as a marker of type I collagen synthesis during statin therapy. We speculate that low adherence to statin therapy might be associated with an increase in heart failure and chronic kidney disease risk among T2DM patients [38,39].

The presence of a positive correlation between BMI and serum TGF-β1 level, a paracrine regulator of extracellular matrix synthesis, supports the consideration of obesity as a myocardial fibrosis accelerator. A significant increase in TIMP1 and ICTP levels has been identified in both groups. Previously, direct relations between plasma TIMP-1 levels and all major CVD risk factors, including male gender, have been demonstrated in the Framingham heart study [40]. It should be noted that obese patients in our study are characterized by an increased HbA1c level, and the higher serum TIMP1 concentration is associated with elevated E/e’ rati, as a marker of LV diastolic dysfunction. In a recent study, TIMP1 has been shown to activate adipogenesis by accelerating lipid accumulation, adipocyte differentiation, and pro-inflammatory cytokine production [41]. Thus, in T2DM patients, TIMP1 may be involved in the target organ damage due to its role in adipogenesis, systemic inflammation, and fibrosis. We suggest this is a major reason why among the numerous factors, TIMP1 is an independent predictor of cardiac fibrosis.

### Study Limitations

The study was performed on a limited non-random patient group who were referred to the Almazov Centre due to poor glycemic control. Both cardiovascular (antihypertensive and hypolipidemic) and antidiabetic therapy were not standardized before patient inclusion. There was a significant overlap of CV risk factors between the groups, and all of them may influence CV remodeling. Myocardial fibrosis was assessed with LGE-MRI imaging, which is less informative when diffuse myocardial fibrosis is present.

## 5. Conclusions

Our cross-sectional study demonstrates that T2DM patients have elevated levels of circulating fibrosis markers and a high prevalence of LGE-MRI. Galectin-3 and TIMP1 serum levels are strongly associated with LGE-MRI in T2DM patients and patients with cardiovascular risk factors. Serum TIMP1 level is an independent predictor of cardiac fibrosis.

## Figures and Tables

**Figure 1 jcm-11-02843-f001:**
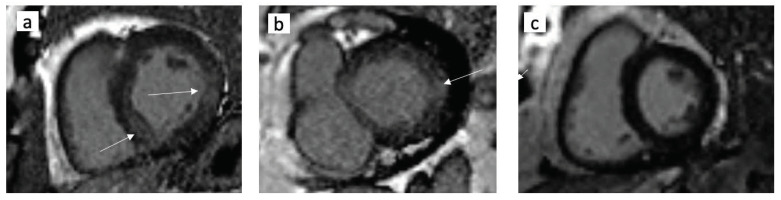
Examples of cardiac MRI with and without LGE: (**a**) a T2DM patient with positive LGE; (**b**) RF patient with positive LGE; (**c**) RF patient without LGE; the white arrows indicate LGE areas in the septum; (**a**) left lateral LV wall (**a**,**b**).

**Table 1 jcm-11-02843-t001:** Baseline characteristics of study subjects.

Variables	T2DM Group *n* = 37	RF Group *n* = 27	HC Group *n* = 15	*p*^1,2^-Value	*p*^2,3^-Value
	1	2	3		
Age, years	57.5 ± 8.4	54.0 ± 8.9	55.6 ± 3.6	0.122	0.378
Male, *n* (%)	17 (46)	12 (44)	7 (47)	0.905	0.735
BMI, kg/m	32.9 ± 6.5	35.6 ± 2.7	23.8 ± 2.0	0.051	<0.001
Waist circumference, cm	109.4 ± 14.0	113.6 ± 8.9		0.186	
Male	111.5 ± 14.3	118.2 ± 8.7		0.166	
Female	107.8 ± 14.0	109.9 ± 7.4		0.598	
T2DM duration, years	9.0 [5.0–12.0]	-			
Hypertension, *n* (%)	21 (57)	19 (70)	0	0.058	
Current smoker, *n* (%)	12 (32)	11 (41)	3 (20)	0.792	0.071
Office systolic BP, mm Hg	131 ± 17	130 ± 17	118 ± 9	0.673	0.002
Office diastolic BP, mm Hg	77 ± 10	81 ± 14	75 ± 8	0.415	0.130
Carotid-femoral PWV, m/s	9.9 ± 2.2	7.9 ± 1.7		0.0002	
Carotid IMT, mm	0.715 ± 0.374	0.618 ± 0.113	0.535 ± 0.114	0.010	<0.001
RHI	1.50 ± 0.35	1.70 ± 0.31		0.019	
eGFR, mL/min/1.73 m^2^	88.4 ± 15.8	90.1 ± 15.9		0.550	
**Echocardiography**					
LA volume index, mL/m^2^	36.7 ± 6.8	32.7 ± 6.0		0.016	
LV mass index, g/m^2^	120.8 ± 32.0	102.0 ± 23.3	90.3 ± 13.4	0.008	0.002
Male	131.9 ± 38.6	111.8 ± 24.1		0.170	
Female	111.3 ± 21.9	93.6 ± 19.7		0.014	
Relative wall thickness	0.448 ± 0.050	0.434 ± 0.048		0.813	
LV EF, %	60.6 ± 5.5	60.9 ± 3.3		0.603	
E/e′	8.2 ± 1.9	7.3 ± 1.2		0.021	
GLS, %	−18.0 ± 3.0	−19.1 ± 2.1		0.110	
**Medication**					
Metformin, *n* (%)	22(59)	1(4)		<0.001	
DPP-4 inhibitors, *n* (%)	5(13.5)	_		_	
Sulphonylureas, *n* (%)	2(5.4)	_		_	
Insulin, *n* (%)	8 (21.6)	_		_	
ACEI or ARB, *n* (%)	21 (56.8)	19 (70)		0.058	
Low-dose aspirin, *n* (%)	13 (48.1)	4 (14.8)		0.002	
Statins, *n* (%)	18 (48.6)	4 (14.8)		<0.01	

Data are presented as mean ± SD or median (interquartile range, IQR) for normal and abnormal distributed continuous variables. Categorical data were expressed as numbers of subjects and percentages. RF—risk factors; HC—healthy control; BMI—body mass index; BP—blood pressure; IMT—intima-media thickness; RHI—reactive hyperemia index; eGFR—estimated glomerular filtration rate (MDRD derived); LA—left atrium; LV—left ventricle; EF—ejection fraction; E/e’—the ratio of mitral inflow early diastolic velocity to the average peak early diastolic mitral annular velocity; GLS—global longitudinal strain; DPP-4—dipeptidylpeptidase-4; ACEI—angiotensin-converting enzyme inhibitor; ARB—angiotensin II receptor blocker; BMI—body mass index; PWV—pulse wave velocity; *p* ^1,2^—comparison between T2DM and RF groups; *p* ^2,3^—comparison with healthy controls.

**Table 2 jcm-11-02843-t002:** Lab tests and fibrosis biomarkers.

Variables	T2DM Group *n* = 37	RF Group *n* = 27	HC Group *n* = 15	*p*^1,2^-Value	*p*^2,3^-Value
	1	2	3		
Total cholesterol, mmol/L	4.84 ± 0.97	5.40 ± 1,11	4.52 ± 1,24	0.056	0.095
HDL-C, mmol/L	1.11 ± 0.26	1.15 ± 0.28	1.16 ± 0.31	0.757	0.62
LDL-C, mmol/L	2.67 ± 0.91	3.49 ± 0.92	2.62 ± 0.98	0.002	0.017
Triglycerides, mmol/L	2.58 ± 1.07	1.88 ± 0.78	1.67 ± 0.93	0.007	0.22
hsCRP, mg/L	2.55 [1.21–4.78]	3.84 [1.99–5.70]	1.67 [0.73–2.96]	0.185	0.11
HbA1c, %	8.9 ± 1.4	5.74 ± 0.85	-	<0.001	-
NT-proBNP, pg/mL	91 [16–148]	27.5 [15.7–47.6]	-	<0.001	-
PICP, ng/mL	136.0 [117.2–166.0]	108.4 [93.2–148.8]	84.0 [69.0–98.3]	0.006	0.001
PIIINP, ng/mL	5.74 [4.43–6.77]	5.09 [4.44–5.96]	3.99 [3.27–4.27]	0.265	0.002
sST2, ng/mL	19.1 [14.9–26.7]	13.2 [10.2–21.8]	12.6 [10.3–16.2]	0.016	0.912
MMP-9, ng/mL	794 [497–1015]	490 [341–911]	277 [253–319]	0.084	0.002
TIMP-1, ng/mL	188 [171–237]	152 [137–185]	141 [120–164]	0.004	0.023
TGF-β1, ng/mL	35.7 [24.5–48.6]	29.6 [15.3–42.2]	12.8 [11.9–18.6]	0.067	<0.001
galectin-3, ng/mL	9.5 [7.8–12.5]	7.8 [6.8–9.9]	6.9 [6.0–7.2]	0.029	0.010
ICTP, ng/mL	5,25 [3.5–6.8]	3.49 [3.03–5.89]	2.98 [2.68–3.97]	0.046	0.030

Data are presented as mean ± SD or median (interquartile range, IQR) for normal and abnormal distributed continuous variables. Categorical data were expressed as numbers of subjects and percentages. HDL-C—high-density lipoproteins; LDL-C—low-density lipoproteins; hsCRP—high-sensitive C-reactive protein; HbA1c—glycated hemoglobin A1c; NT-proBNP; RF—risk factors; HC—healthy control; PICP—carboxy-terminal propeptide of collagen 1; PIIINP—amino-terminal propeptide of collagen 3; sST2—soluble suppression of tumorigenicity 2; MMP-9—matrix metalloproteinase 9; TIMP-1—tissue inhibitor of metalloproteinase 1; TGF-β1—transforming growth factor β-1; ICTP—carboxy-terminal telopeptide of collagen 1. *p*
^1,2^ and *p*
^2,3^—compression between groups.

**Table 3 jcm-11-02843-t003:** Univariate logistic regression analysis of factors associated with cardiac fibrosis as detected by MRI.

	Estimate	Standard Error	Wald Stat.	Lower CL—95. %	Upper CL—95. %	*p*
	**All subjects (T2DM+RF+HC groups)**					
T2DM: Yes	1.101	0.32	11.807	0.473	1.728	<0.001
BMI, kg/m^2^	−0.104	0.053	3.832	−0.209	0.0001	0.05
Metformin baseline therapy: Yes	1.06	0.305	12.042	0.461	1.659	<0.001
Statins: Yes	0.785	0.281	7.794	0.234	1.335	0.005
PWV, m/s	0.327	0.135	5.898	0.063	0.591	0.015
RHI	−1.398	0.8	3.053	−2.966	0.17	0.081
LV hypertrophy	0.799	0.299	7.161	0.214	1.384	0.008
HbA1c, %	0.52	0.167	9.682	0.192	0.848	0.002
LDL-C, mM/L	−0.534	0.294	3.306	−1.109	0.042	0.069
TIMP-1, ng/mL	0.018	0.006	8.438	0.006	0.03	0.004
Galectin-3, ng/mL	0.225	0.09	6.159	0.047	0.403	0.013
	**T2DM patients only**					
PWV, m/s	−0.351	0.194	3.261	−0.73	0.03	0.042
TIMP-1, ng/mL	−0.02	0.008	5.187	0.036	−0.003	0.05

**Table 4 jcm-11-02843-t004:** General multivariate regression model of cardiac fibrosis predictors (as detected by MRI).

	Estimate	Standard Error	Wald Stat.	Lower CL—95, %	Upper CL—95, %	*p*
	**All subjects (T2DM+RF+HC groups)**					
Intercept	−5.596	2.189	6.533	−9.887	−1.304	0.01
PWV, m/s	0.12	0.175	0.471	−0.223	0.464	0.492
TIMP-1, ng/mL	0.014	0.007	4.042	0.0003	0.028	0.044
Galectin-3, ng/mL	0.136	0.109	1.57	−0.077	0.349	0.21
T2DM: Yes	0.67	0.426	2.466	−0.166	1.506	0.116
LV hypertrophy: Yes	0.524	0.412	1.623	−0.282	1.331	0.203
	**T2DM patients only**					
Intercept	6.607	2.778	5.657	1.163	12.052	0.02
PWV, m/s	−0.353	0.218	2.623	−0.780	0.074	0.12
TIMP-1, ng/mL	−0.018	0.009	4.596	−0.035	−0.002	0.03

**Table 5 jcm-11-02843-t005:** Multivariate model with additional factors only that predicted cardiac fibrosis in all subjects (as detected by MRI).

	Estimate	Standard Error	Wald Stat.	Lower CL—95, %	Upper CL—95, %	*p*
	**All subjects (T2DM+RF+HC groups)**					
Intercept	−7.128	1.996	12.749	−11.04	−3.215	0.0004
PWV, m/s	0.208	0.155	1.796	−0.096	0.512	0.18
TIMP-1, ng/mL	0.017	0.007	6.265	0.004	0.029	0.01
Galectin-3, ng/mL	0.189	0.099	3.648	−0.005	0.384	0.049

## Data Availability

All relevant data are included in the manuscript.

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
