# Peer review of "Simple Predictors for Cardiac Fibrosis in Patients with Type 2 Diabetes Mellitus: The Role of Circulating Biomarkers and Pulse Wave Velocity"

_jcm, 2022, doi:10.3390/jcm11102843_

Round 1

Reviewer 1 Report

T2DM is a multifactorial disease with many comorbidities.

There is a significant overlap of CV risk factors between T2DM and RF group and all of them (obesity, hypertension, chronic inflammation, hyperglycemia) may influence cardiovascular remodeling.

For future investigations patients in the investigated groups should be performed in pares with comparable CV and anthropometric factors.

Author Response

Dear Reviewers,

We thank you and the Editor for the careful evaluation of our manuscript and the opportunity to provide its revised version. We have addressed all the comments and suggestions, as you will realize from our point-by-point responses below.

After implementation of all suggestions, which we found very useful, we do believe that the manuscript has improved significantly.

We do hope that you will find the changes satisfactory and the manuscript suitable for publication in the Journal of Clinical Medicine.

With kind regards,

the authors.

Response to Reviewer 1

  1. Comment: English language and style are fine/minor spell check required

Response: According to the reviewer suggestion, we have checked the English language and style and corrected all found mistakes.

  1. Comment: For future investigations patients in the investigated groups should be performed in pares with comparable CV and anthropometric factors.

Response: We thank the reviewer and will address the suggested point in our future studies.

Reviewer 2 Report

Luneva et al aimed in this study to identify simple predictors of cardiac fibrosis in patients with type 2 diabetes mellitus based on the analysis of circulating fibrosis biomarkers and arterial stiffness. Therefore they included patients with T2DM and cardiovascular risk factors as well as healthy people who underwent echocardiography, cardiac MRI, PWV, RHI peripheral arterial tonometry, carotid ultrasonography and assessment of serum fibrosis biomarkers.

Although the manuscript is well written and well structured, the manuscript has methodological flaws and it is not clear why these various patient groups were investigated and what is the primary and secondary aim of this trial. The study was not registered in a database before the beginning of the trial.

Please explain why patients with CV risk factors and healthy people were included in your trial, even tough the aim of your study is to identify predictors of cadiac fibrosis in patients with T2DM?Please also adapt the introduction section why these cohorts were investigated in this trial.

Are there any secondary aims of the trial which justifies the inclusion of the above participants?

Was the study registered before start? e.g. clinicaltrials.gov 

Line 176: please "not significant" instead of "insignificant"

Line 180: pulse wave velocity

Table 1 Line 200:

P2,3 comparison between RF group and healthy controls

Please add BMI, PWV also to the abbreviations section 

Table 2:

Please explain MMP-9, TMIP-1, TGF-beta1, ICTP ,....

Conclusion:

Your conclusion does not fit with research question of your trial, because it also includes patient with risk factors.

Author Response

Dear Reviewer,

We thank You and the Editor for the careful evaluation of our manuscript and the opportunity to provide its revised version. We have addressed all the comments and suggestions, as you will realize from our point-by-point responses below.

After implementation of all suggestions, which we found very useful, we do believe that the manuscript has improved significantly.

We do hope that you will find the changes satisfactory and the manuscript suitable for publication in the Journal of Clinical Medicine.

With kind regards,

the authors.

Response to Reviewer 2

1. Comment: Although the manuscript is well written and well structured, the manuscript has methodological flaws and it is not clear why these various patient groups were investigated and what is the primary and secondary aim of this trial.

Response: Thank you for this important comment. We have added the rationale for the inclusion of comparison groups, right below the study aim:

“T2DM has been shown associated with tissue fibrosis in general and cardiac fibrosis in particular [12]. Plasma concentrations of circulating biomarkers, that may characterize the presence and extent of fibrosis, is associated with other morbidity and risk factors, such as obesity and hypertension. Moreover, their reference level should be evaluated in healthy subjects, for the assessment of their significance, when changed. Therefore, along with T2DM patients we included two more subgroups: subjects without T2DM but with cardiovascular risk factors, and healthy controls.” One more reference has been cited in this text, and added to the reference list.

Additionally, we have added more detailed description of primary and secondary study measures (2.1 sub-section):

“The primary study assessment measure included the evaluation of a possible association between cardiac fibrosis as detected by CMR, artery stiffness, and circulating biomarkers. The secondary study analysis was the evaluation of factors independently associated with the presence of cardiac fibrosis.”

2. Comment: The study was not registered in a database before the beginning of the trial.

Response: Indeed, the study was not registered in any public database (such as clinicaltrials.gov), since our local country regulations do not require a special registration of observational studies. However, the study was registered as a section of a general project “Major research project – Development of technologies for the prevention and treatment of heart failure based on neuromodulation”. This projects received registration on a protected website for government use and expert analysis (https://sstp.ru/, restricted login with no public access). We have added a statement regarding this registration at the end of 2.1 sub-section:

“This observational study was registered as a part of an umbrella project #075-15-2020-800 by the Ministry of Science and Higher Education. The local legislation does not require observational studies registration in the public databases.”

3. Comment: Please explain why patients with CV risk factors and healthy people were included in your trial, even tough the aim of your study is to identify predictors of cadiac fibrosis in patients with T2DM? Please also adapt the introduction section why these cohorts were investigated in this trial. Are there any secondary aims of the trial which justifies the inclusion of the above participants?

Response: Since incident and unexpected myocardial fibrosis may eventually be detected in patients with cardiovascular diseases and even risk factors only (we based our assumption on the following studies: doi: 10.1161/CIRCIMAGING.113.001768; 10.1093/europace/euab167; 10.1161/CIRCULATIONAHA.117.032175), we have included two additional groups of subjects for the comparison of biomarkers’ levels and MRI findings: patients with CV risk factors and healthy subjects. It is suggested that plasma concentrations of circulating biomarkers, that may characterize the presence and extent of fibrosis, may be associated with other morbidity and risk factors, such as obesity and hypertension. Moreover, their reference level should be evaluated in healthy subjects, for the assessment of their significance, when changed. Therefore, along with T2DM patients we included two more subgroups into the study: subjects without T2DM but with cardiovascular risk factors, and healthy controls.

We have added the following text to the Introduction clarifying the inclusion of two comparison groups: “T2DM has been shown associated with tissue fibrosis in general and cardiac fibrosis in particular [12]. Plasma concentrations of circulating biomarkers, that may characterize the presence and extent of fibrosis, is associated with other morbidity and risk factors, such as obesity and hypertension. Moreover, their reference level should be evaluated in healthy subjects, for the assessment of their significance, when changed. Therefore, along with T2DM patients we included two more subgroups: subjects without T2DM but with cardiovascular risk factors, and healthy controls.” One more reference has been cited in this text, and added to the reference list.

4. Comment: Was the study registered before start? e.g. clinicaltrials.gov 

Response: Indeed, the study was not registered in any public database (such as clinicaltrials.gov), since our local country regulations do not require a special registration of observational studies. However, the study was registered as a section of a general project “Major research project – Development of technologies for the prevention and treatment of heart failure based on neuromodulation”. This projects received registration on a protected website for government use and expert analysis (https://sstp.ru/, restricted login with no public access). We have added a statement regarding this registration at the end of 2.1 sub-section:

“This observational study was registered as a part of an umbrella project #075-15-2020-800 by the Ministry of Science and Higher Education. The local legislation does not require obresvational studies registration in public databases.” The study was approved by a local ethics committee.

5. Comment: Line 176: please "not significant" instead of "insignificant"

Response:  Corrected

6. Comment: Line 180: pulse wave velocity

Response: Corrected

7. Comment: Table 1 Line 200:

Response: Corrected

8. Comment: P2,3 comparison between RF group and healthy controls

Response: Corrected

9. Comment: Please add BMI, PWV also to the abbreviations section 

Response: Corrected

10. Comment: Table 2: Please explain MMP-9, TMIP-1, TGF-beta1, ICTP ,....

Response: Corrected

11. Comment: Conclusion: Your conclusion does not fit with research question of your trial, because it also includes patient with risk factors.

Response: We thank the reviewer for this particular very important comment. Indeed, it was a shortcoming of our analysis. We have now performed additional regression analysis of demographic, clinical and laboratory parameters associated with the presence of late gadolinium enhancement on MRI for the main group of patients (patients with type 2 diabetes mellitus). The results of this analysis clearly demonstrated that TIMP-1 and pulse wave velocity (PWV) are again associated with cardiac fibrosis, and TIMP-1 level is an independent predictor of cardiac fibrosis in the multivariate analysis. According to this addition, we amended the Methods (2.9 – Statistical analysis: “The regression analysis was performed for the total study population and for T2DM patients separately.”) and Results sections (3.3. Echocardiography and cardiac MRI analysis – “In T2DM patients the regression analysis confirmed significant associations of TIMP-1 and PWV with cardiac fibrosis, and the multivariate model identified TIMP-1 as the only factor independently associated with LGE positive findings (tables 3, 4).”). We have also added the results of regression analyses in the T2DM group to tables 3 and 4.

Therefore, according to the findings, the conclusion may now sound even stronger regarding the T2DM subjects. We have left the Conclusions the same as in the previous manuscript version: “Our cross-sectional study demonstrates that T2DM patients have elevated levels of circulating fibrosis markers and a high prevalence of LGE-MRI. Galectin-3 and TIMP1 serum levels are strongly associated with LGE-MRI in T2DM patients and patients with cardiovascular risk factors. Serum TIMP1 level is an independent predictor of cardiac fibrosis.” We hope that you will find this addition satisfactory.

Reviewer 3 Report

Major concerns:

  1. Page 2 and line 69: The inclusion criteria for the T2D group: Hypertension or dyslipidemia. The levels of blood pressure and LDL-C should be indicated in the text.
  2. Page 2 and line 85: what does "48C" refer to?
  3. Please list names of vendors, cities, and countries for all supplies in Materials and Methods.
  4. Page 3 and line 122: what does "CMR" refer to?
  5. Page 4 and line 147. One way ANOVA and post-hoc should be used for statistical analysis of groups more than 2.  
  6. Table 1. T2DM group (n=37). The calculation of % for Hypertension and Medication groups was inaccurate.
  7. Table 2. LDL-C was 2.67+/-0.91 (T2DM group) vs. 3.49+/-0.92 (RF group) and P value was 0.002 (highly significant difference). LDL-C was 2.62+/-0.98 (HC group) vs. 3.49+/-0.92 (RF group) and P value was 0.17 (no difference). Can authors recheck these results?
  8. Figure 1. Distributions of LGE in three different patients with T2DM. It is recommended to show representatives from three groups (T2D, RF, and HC). 
  9.  Minors: spelling errors and grammar errors should be corrected.

Author Response

Dear Reviewer,

We thank You and the Editor for the careful evaluation of our manuscript and the opportunity to provide its revised version. We have addressed all the comments and suggestions, as you will realize from our point-by-point responses below.

After implementation of all suggestions, which we found very useful, we do believe that the manuscript has improved significantly.

We do hope that you will find the changes satisfactory and the manuscript suitable for publication in the Journal of Clinical Medicine.

With kind regards,

the authors.

Response to Reviewer 3

1. Comment: English language and style are fine/minor spell check required

Response: According to the reviewer suggestion, we have checked the English language and style and corrected all found mistakes.

2. Comment: Page 2 and line 69: The inclusion criteria for the T2D group: Hypertension or dyslipidemia. The levels of blood pressure and LDL-C should be indicated in the text.

Response: Thank you. We have added this information: “The RF group inclusion criteria were the combination of two common risk factors: obesity (BMI>30.0 kg/m2) and hypertension (office blood pressure level > 140/90 mm Hg) or dyslipidemia (the history of LDL cholesterol > 3 mmol/l).”

3. Comment: Page 2 and line 85: what does "48C" refer to?

Response: Corrected to “4°C”.

4. Comment: Please list names of vendors, cities, and countries for all supplies in Materials and Methods.

Response: We have added this information to each material listed.

5. Comment: Page 3 and line 122: what does "CMR" refer to?

Response: Thank you. We have changed this abbreviation to “cardiac magnetic resonance imaging (cardiac MRI)” through the whole text.

6. Comment: Page 4 and line 147. One way ANOVA and post-hoc should be used for statistical analysis of groups more than 2.  

Response: Thank you. Indeed, this analysis was used but not listed in the Statistical analysis section. We have added this information to sub-section 2.9. Statistical analysis.

7. Comment: Table 1. T2DM group (n=37). The calculation of % for Hypertension and Medication groups was inaccurate.

Response: Thank you. Indeed, there was a systematic mistake in percentage calculations. We have now corrected the percentages in the table.

8. Comment: Table 2. LDL-C was 2.67+/-0.91 (T2DM group) vs. 3.49+/-0.92 (RF group) and P value was 0.002 (highly significant difference). LDL-C was 2.62+/-0.98 (HC group) vs. 3.49+/-0.92 (RF group) and P value was 0.17 (no difference). Can authors recheck these results?

Response: Thank you. There was a missed zero. Now corrected to 0.017.

9. Comment: Figure 1. Distributions of LGE in three different patients with T2DM. It is recommended to show representatives from three groups (T2D, RF, and HC). 

Response: We have replaced with new figures obtained from different groups.

10. Minor comment: spelling errors and grammar errors should be corrected.

Response: We have revised the manuscript and corrected all found spelling and grammar errors.

Round 2

Reviewer 2 Report

many thanks for the corrections!

Author Response

Dear Reviewer,

Thank you for the thorough analysis of our manuscript and overall positive conclusion.

With kind regards,

the authors.

Reviewer 3 Report

This is a revised manuscript. Authors have addressed all concerns. The quality of this manuscript has been improved.

New comments:

  1. The images in Figure 1 are fuzzy. The figure legend should include the arrows in the images.
  2. Page 4 and Line 176, please describe the method for post-hoc analysis of ANOVA.  

Author Response

Dear Reviewer,

Enclosed please find our revised manuscript entitled “Simple Predictors for Cardiac Fibrosis in Patients with Type 2 Diabetes Mellitus: the Role of Circulating Biomarkers and Pulse Wave Velocity" by E.Luneva and co-workers. After implementation of all considerations raised by the referees, we would like to resubmit this Original Research Paper.

Specifically, we have added information regarding ANOVA post hoc analysis (the Tukey-Kramer test added to line 179-180, page 4), quality of the figure 1 was improved and arrows’ description added to the figure legend. All new modifications are marked by the blue color. Please find below our point-by-point response to you comments.

We thank you for the thorough evaluation of our manuscript.

With kind regards,

the authors.

++++++++++

Comment. This is a revised manuscript. Authors have addressed all concerns. The quality of this manuscript has been improved.

Response: Thank you very much for the valuable comments and suggestions.

++++++++++

Comment: New comments: The images in Figure 1 are fuzzy. The figure legend should include the arrows in the images.

Response: We have updated the figure with better quality images. The figure legend was amended, and we added the description of the arrows and what they point on.

++++++++++

Comment: Page 4 and Line 176, please describe the method for post-hoc analysis of ANOVA. 

Response: We have added the following information within brackets: “The one-way ANOVA and post-hoc (Tukey-Kramer test) were also used for comparison of parameters in three groups.”

This manuscript is a resubmission of an earlier submission. The following is a list of the peer review reports and author responses from that submission.